# UPLC-ESI-MS/MS-Based Analysis of Various Edible *Rosa* Fruits Concerning Secondary Metabolites and Evaluation of Their Antioxidant Activities

**DOI:** 10.3390/foods13050796

**Published:** 2024-03-04

**Authors:** Ming Ni, Junlei Chen, Mao Fu, Huanyang Li, Shengqian Bu, Xiaojiang Hao, Wei Gu

**Affiliations:** 1School of Pharmaceutical Sciences, Guizhou University, Guiyang 550014, China; niming8883@163.com; 2State Key Laboratory of Functions and Applications of Medicinal Plants, Guizhou Medical University, Guiyang 550014, China; stefanpin@163.com (J.C.); fm33557@163.com (M.F.); 19308503824@163.com (H.L.); bushengqian@163.com (S.B.); 3Natural Products Research Center of Guizhou Province, Guiyang 550014, China; 4State Key Laboratory of Phytochemistry and Plant Resources in West China, Kunming Institute of Botany, Chinese Academy of Sciences, Kunming 650201, China

**Keywords:** *Rosa roxburghii*, *Rosa laevigata*, *Rosa sericea*, metabolic profiling, antioxidant activity

## Abstract

The genus *Rosa* is globally popular with well-established applications since it has a high edible and medicinal value. However, relatively limited research has been conducted on the composition and quality of wild *Rosa* fruits. The present study aimed to compare the properties and chemical components of five wild edible *Rosa* fruits, *Rosa roxburghii*, *Rosa sterilis*, *Rosa laevigata*, *Rosa davurica*, and *Rosa sericea*. The UPLC-ESI-MS/MS approach identified the key metabolites among the five *Rosa* fruits as flavonoids, phenolic acids, and organic acids. The main differential metabolites among the five fruits are flavonoids (22.29–45.13%), phenolic acids (17–22.27%), and terpenoids (7.7–24%), respectively. In total, 125 compounds served as potential markers for the five *Rosa* species. Differential metabolic pathways of five *Rosa* fruits were analyzed using the KEGG approach. *Rosa laevigata* fruits showed the highest total polysaccharide (TPS) content of 64.48 g/100 g. All the five *Rosa* extracts effectively decreased the levels of malondialdehyde while increasing the activities of superoxide dismutase and glutathione peroxidase in the H_2_O_2_-induced HaCaT cell model, demonstrating high potential for antioxidant development. Our findings suggest that the five studied *Rosa* fruits exhibit biological activity and edible value worth further exploration.

## 1. Introduction

Recently, the research and development of nutraceuticals have progressed dramatically. In this field, functional juice and beverage products have attracted considerable attention, owing to the great diversity of available vegetables and fruits. One such unexploited nutritional source is rose hip (*Rosa* fruits) [1]. The genus *Rosa* is one of the most widespread members of the *Rosaceae* family, with more than 200 species being found in the temperate and subtropical zones of the Northern Hemisphere [2]. Roses have been widely acclaimed aesthetically and economically for their exquisite blooms and spectacular growth and are popularized worldwide for their nutritious, therapeutic, ornamental, and cosmetic usefulness. Many foodstuffs, such as jams and jellies, and drinks, including tea and alcoholic beverages are prepared from rose hips [3]. In addition, *Rosa* species are traditionally used in Asia and Europe to treat various diseases, including those of the liver, kidney, lungs, heart, and stomach [2].

*Rosa roxburghii* Tratt. is native to China and is mainly distributed in the mountainous areas at an altitude of 1000–1600 m, particularly in the karst areas of Guizhou province [4]. The cultivation area of *R. roxburghii* has expanded and is estimated to exceed 140,000 ha in Guizhou province by 2021, and the output value of *R. roxburghii* fruit (RRT-F) is CNY 520 million (http://www.chinanews.com.cn/cj/2022/01-02/9642987.shtml, accessed on 2 January 2022). RRT-F is popular for the highest contents of vitamin C (Vc) and superoxide dismutase (SOD) among other common fruits [5]. Modern pharmacological studies have shown that RRT-F has antioxidant, antimutagenic, anti-atherogenic, and antitumor effects, along with genoprotective and radioprotective activities [6]. *Rosa sterilis* S. D. Shi was first discovered 30 years ago in Anshun, Guizhou province, China; it was explored by Shengde Shi in 1985 [6]. Previous studies based on ITS sequence analysis proposed that *R. sterilis* is a natural hybrid of *R. roxburghii* and *R. longicuspis* [7]. However, the latest comparative genomics research results indicated that *R. sterilis* and *R. roxburghii* are two independent species, and there is a closer genetic relationship between *R. sterilis* and *R. kweichowensis* [8]. As a novel fruit resource, *R. sterilis* fruit (RSS-F) exhibited high nutritional value and a broad market prospect [9]. The components of *R. sterilis* showed antioxidant, immunity-promoting, anticancer, and anti-aging effects [10,11]. *Rosa laevigata* Michx., unique to China, commonly known as the “Jin-Ying-Zi” in Chinese, is an evergreen climbing shrub prevalent in the southern regions of China. Its fruits were documented in the Chinese Pharmacopoeia and are mainly used for the treatment of nocturnal emission, nocturnal enuresis, frequent urination, metrorrhagia, and diarrhea [12]. Moreover, as a recognized functional and health food in Asian countries, the fruit of *R. laevigata* (RLM-F) has been developed as a third-generation wild fruit by the Ministry of National Health of China [13,14]. *Rosa davurica* Pall., a deciduous shrub, is mainly distributed in northeastern China, Korea, Japan, southeastern Siberia, and eastern Asia. *R. davurica* fruit (RDP-F) is generally used as a traditional medicine in many countries such as China, Korea, and Japan [15,16,17]; however, only in China is it widely used in food. The health-promoting effects of *R. davurica*, such as its antioxidant, anti-inflammatory, antiviral, and hypoglycemic activities, are well-established [17,18,19,20]. *R. davurica* fruit (RDP-F) is also commonly consumed as a beverage and health food in China [21]. *Rosa sericea* Lindl., commonly known as the silky rose, is mainly distributed in southwestern China {Guizhou, Sichuan, Xizang, and Yunnan} and India, Bhutan, Nepal, and Myanmar [22]. The fruit and root of *R. sericea*, also called “Shancili,” have been used as traditional medicine to treat abdominal distention, diarrhea, chronic dysentery, menorrhagia, and metrorrhagia in China [23]. *R. sericea* fruit (RSL-F) is also consumed in southwestern China as a foodstuff; it is nutrient-rich and exhibits great potential as a health food.

A characteristic feature of many fruits is their exceptional antioxidant capacity. The potential of *Rosa* plants deserves particular attention, considering the growing demand of preventing and resisting factors such as aging caused by oxidation. Recently, roses have gained considerable attention and are being used in cosmetics. Therefore, this study aimed to further explore new resources with antioxidant potential in the *Rosa* genus.

With the recent advent of metabolomics, high-performance liquid chromatography (HPLC)–mass spectrometry coupled with multivariate data analysis has been applied to analyze metabolite profiles and successfully detect variations in the composition of phytomedicines, foods, and biofluids [24,25]. Recently, widely targeted metabolomics using ultra-high-performance liquid chromatography–electrospray ionization tandem mass spectrometry (UPLC-ESI-MS/MS) is being preferred for the analysis and identification of plant metabolites because of its rapid separation, high sensitivity, and wide coverage [26]. Although omics studies have recently been conducted on the economically popular plant *R. roxburghii* [27], the edible fruits of other wild *Rosa* species have been disregarded. Thus, we hypothesize the importance of comprehensively studying and comparing the properties and metabolites of some typical edible *Rosa* fruits from the perspectives of taste and development.

In this study, the secondary metabolite composition of the five mentioned edible *Rosa* fruits from China was analyzed and compared using widely targeted metabolomics, and differentially expressed metabolites among the species were identified.

## 2. Materials and Methods

### 2.1. Chemicals and Reagents

DMEM (Dulbecco’s Modified Eagle Medium) and FBS (Fetal Bovine Serum) were purchased from Hyclone (Shanghai, China). HaCaT (human immortalized keratinocyte) cells were from Jarvis (Wuhan, China) Biological Pharmaceutical Co., Ltd. (Wuhan, China). Assay kits for detecting glutathione (GSH) and malondialdehyde (MDA) were purchased from Nanjing Jiancheng Biology (Nanjing, China). All the other chemicals were of analytical grade. All other solvents used for HPLC analysis were of chromatography grade.

### 2.2. Plant Materials

Five *Rosa* samples, including the fruits of *R. roxburghii* (RRT-F, GC_1), *R. sterilis* (RSS-F, GC_2), *R. laevigata* (RLM-F, GC_3), *R. davurica* (RDP-F, GC_4), and *R. sericea* (RSL-F, GC_5) were collected from the Guiyang city of Guizhou province, Majiang county of Guizhou province, Kaili city of Guizhou province, Bijie city of Guizhou province, and Daxinganling city of Heilongjiang province during the period from June to November 2022. (Figure 1). Voucher specimens were identified by Dr. Wei Gu and deposited at the Natural Products Research Center of Guizhou Province. The voucher specimens number were GZCNG-2022-0078, GZCNG-2022-0189, GZCNG-2022-0102, GZCNG-2022-0157, GZCNG-2022-0013, respectively. The fruits were dried at 50 °C and crushed using a high-speed disintegrator (Wenling Auari Traditional Chinese Medicine Machinery Co., Ltd., Wenling, China). The sample powders were sealed and stored in an ultra-low-temperature refrigerator at −80 °C for a maximum of four weeks.

### 2.3. Metabolites Extraction

The extraction process was based on a previously reported method [28]. The ground powder (500 g) was extracted with 80% ethanol (3 × 200 mL) under reflux for 4 h and filtered using a cotton plug followed by medium-speed qualitative filter paper (9 cm, Solarbio, Beijing, China). The filtrate was concentrated under reduced pressure (55 °C) to obtain the RRT-F (37.8 g), RSS-F (40.2 g), RLM-F (49.8 g), RDP-F extracts (39.7 g), and RSL-F extracts (38.6 g). The extracts were freeze-dried in a vacuum freeze dryer (Scientz-100F, Ningbo Scientz BIOTECH. CO., LTD., Ningbo, China, 1 pa, −78 °C, 2 h) and crushed using a mixer mill (MM 400, Retsch, Haan, Germany) with zirconia beads for 1.5 min at 30 Hz. The lyophilized powder (100 mg) was dissolved in 1.2 mL 70% methanol and vortexed. The samples were placed in a refrigerator at 4 °C for 12 h, then centrifuged at 1000× *g* for 10 min, and the supernatant filtered using a nylon needle filter (SCAA-104, 0.22 μm pore size; ANPEL, Shanghai, China) before UPLC-MS/MS analysis.

### 2.4. Non-Targeted Metabolite Analysis Using UPLC-ESI-MS/MS

The samples were analyzed using a UPLC-ESI-MS/MS system (UPLC, Shim-pack UFLC SHIMADZU CBM A system; MS, QTRAP^®^ 4500+ System). The UPLC system was equipped with a Waters ACQUITY UPLC HSS T3 C18 column (1.7 µm, 2.1 mm × 100 mm). The mobile phase consisted of water/formic acid (99.9:0.1 *v*/*v*, solvent A) and acetonitrile/formic acid (99.9:0.1 *v*/*v*; solvent B). The elution gradient was as follows: 0–10 min, 5–95% B; 10–11 min, 95% B; 11–11.1 min, 95–5% B; 11.1–15 min, 5% B. The flow rate was 0.4 mL/min, the column temperature was 40 °C, and the injection volume was 2 μL.

Linear ion trap (LIT) and triple quadrupole (QQQ) scans were acquired using a QQQ-LIT mass spectrometer (Q TRAP), AB4500 Q TRAP UPLC/MS/MS system, equipped with an ESI Turbo Ion–Spray interface, operating in positive and negative ion modes, and controlled using the Analyst 1.6.3 software (AB Sciex). The ESI source operation parameters were as follows: ion source, turbo spray; source temperature, 550 °C; ion spray voltage, (positive ion mode) 5500 V/(negative ion mode)-4500 V; ion source gas I, gas II, and curtain gas were set at 50, 60, and 25.0 psi, respectively; and the collision-activated dissociation was high. Instrument tuning and mass calibration were performed with 10 and 100 μmol/L polypropylene glycol solutions in QQQ and LIT modes, respectively. QQQ scans were acquired in the multiple reaction monitoring (MRM) experiments with a collision gas (nitrogen) in the medium. The declustering potential (DP) and collision energy (CE) for individual MRM transitions were performed and further optimized. A specific set of MRM transitions was monitored for each period according to the metabolites eluted within this period. The qualitative analysis of substances was based on self-built databases and secondary spectral information. Metabolite quantification is achieved through multi-reaction monitoring (MRM) mode analysis using triple quadrupole mass spectrometry. Mass spectrometry data processing was conducted using Analyst 1.6.3 software (AB Sciex).

### 2.5. Multivariate Statistical Analysis

The edited data matrix was imported into the MetaboAnalystR packages in the R software (v 1.22.0) platform for multivariate analysis, including principal component analysis (PCA) and orthogonal partial least squares discriminant analysis (OPLS-DA). We analyzed the metabolomic data based on the OPLS-DA model, drew score maps for each group, and further demonstrated differences between the groups. Based on the results of OPLS-DA, the variable importance in projection (VIP) of the multivariate analysis was obtained. This experiment adopted a combination of fold-change (FC) and VIP values from the OPLS-DA model to screen for differential metabolites, which were annotated and displayed in the Kyoto Encyclopedia of Genes and Genomes (KEGG) database [29].

### 2.6. Determination of Total Polysaccharide (TPS) Content

The TPS content in five *Rosa* fruits was detected using the phenol–sulfuric acid method, with minor modifications [30]. Briefly, the samples were dried at 50 °C and ground into a powder. An amount of 2.0 g of the ground fruits was extracted in hot water (1:25, *w*/*v*) at 100 °C for 3 h. After cooling, the weight was replenished with purified water. The extract solution (2 mL) was diluted in a 100 mL volumetric flask with ultrapure water. A standard glucose solution was prepared at a concentration of 0.05642 mg/mL (dried to constant weight before use); following this, 60, 120, 180, 240, 300, 360, and 400 μL of glucose solution was taken in glass test tubes. The volume was adjusted to 400 μL with pure water. Further, 200 μL of 5% phenol (*w*/*v*) and 1 mL sulfuric acid were added. The tubes were placed in a water bath at 100 °C for 15 min and then placed in a refrigerator at 4 °C. Glucose content was measured using an enzyme-labeling instrument at a wavelength of 490 nm.

The sample solution (20 mL) was then diluted to 100 mL ultrapure water. The reaction was performed in test tubes with a ratio of 2:1:5 among the sample solution, 5% phenol (*w*/*v*), and sulfuric acid in a hot water bath at 100 °C for 15 min. After cooling at 20 °C, the absorbance was measured using an MULTISKAN MK3 microplate reader (Thermo Fisher Scientific, Waltham, MA, USA) at a wavelength of 490 nm. The TPS content was calculated from the absorbance and standard curve.

### 2.7. Determination of Antioxidant Activity

#### 2.7.1. Determining the Effects of *Rosa* Samples on H_2_O_2_-Damaged HaCaT Cells

The effects of the samples on human immortalized epidermal (HaCaT) cell viability were determined using the cell counting Kit-8 (CCK-8) method [31]. HaCaT cells were cultured in Dulbecco’s Modified Eagle Medium supplemented with 10% (*v*/*v*) fetal bovine serum, streptomycin, and penicillin. Cells in the logarithmic phase were seeded in 96-well plates (1 × 10^4^ cells/well) and incubated in an atmosphere of 5% (*v*/*v*) CO_2_. The culture temperature was set at 37 °C. Sample solutions (100 μL) at different concentrations were added into each well and cultivated for 24 h. Following this, 10 μL of CCK-8 was added. Cell viability was assayed 2 h later by monitoring the absorbance at 450 nm and was computed according to Formula (1):Cell viability (%) = (A_sample_ − A_blank_)/(A_control_ − A_blank_) × 100,(1)

To explore the oxidative protective effects, non-toxic concentrations of *Rosa* samples (0.25 mg/mL) were tested using the CCK-8 method. Briefly, after *Rosa* samples and hydrogen peroxide (H_2_O_2_, 2 mg/mL) treatment, CCK-8 (10 μL) was adjected in the corresponding well at 37 °C for 2 h. The optical density was measured at 540 nm using a MULTISKAN MK3 microplate reader (Thermo Fisher Scientific, Waltham, MA, USA). HaCaT cell viability was expressed as a percentage of the control cell group. V_C_ was used as a positive control.

#### 2.7.2. Determination of Superoxide Dismutase (SOD), Glutathione Peroxidase (GSH-Px) Activity, and Malondialdehyde (MDA) Content

After grouping, 2 mg/mL of H_2_O_2_ diluted with the culture medium was added to induce cell damage for 4 h, followed by repeated freezing and thawing until the cell membranes ruptured, and the test solutions were collected. SOD and GSH-Px activities and MDA content were measured according to the manufacturer’s instructions [32]. Each experiment was repeated three times.

## 3. Results

### 3.1. Metabolites Identification

Widely targeted metabolomic analysis of the five *Rosa* fruit samples was performed using UPLC-ESI-MS/MS. The peak areas of the metabolites, representing the relative contents of the corresponding metabolites, were calculated and corrected using the MultiaQuant software (3.0) (Appendix A). Compared with the negative ESI mode, positive ionization showed more definite peaks for all samples. A total of 514 metabolites, including 229 compounds from the positive mode and 285 compounds from the negative mode of the mass spectra, were tentatively identified by comparing the MS spectrum with in-house and public metabolite databases (Appendix A). These metabolites mainly include several types of compounds such as flavonoids, phenolic acids, organic acids, terpenoids, alkaloids, lignans, etc. According to previous reports, flavonoids and phenolic acids are known to be the predominant components in *Rosa* samples [33,34]. Rose hips (*Rosa* L.), including *Rosa persica*, *Rosa spinosissima*, *Rosa platyacantha*, *Rosa beggeriana*, *Rosa iliensis*, *Rosa laxa*, *Rosa laxa* var. *Tomurensis*, *Rosa acicularis*, and *Rosa* ‘Tianshan Xiangyun’ were studied; the results show that flavonoids and phenolic compounds accounted for more than 50% of all the metabolites in the five *Rosa* samples [35].

Among these, a total of 471 compounds were identified from RRT-F which mainly comprised flavonoids (34.04%), phenolic acids (18.08%), organic acids (16.17%), terpenoids (11.38%), alkaloids (4.89%), etc., and the five compounds with the highest relative content were histidinol (3.04%), citraconic acid (2.38%), hesperetin 5-O-glucoside (2.04%), spiraeoside (2.05%), and (Rs)-mevalonic acid (1.95%). A total of 467 compounds were detected in RSS-F, mainly including terpenoids (22.62%), flavonoids (22.35%), organic acids (15.95%), phenolic acids (14.33%), and alkaloids (9.37%). The five compounds with the highest content in RSS-F were ergotamine (4.86%), euscaphic acid (4.21%), histidinol (3.85%), 2,3-dihydroxybenzoic acid (3.10%), and protocatechuic acid (3.06%). There were 428 compounds identified from RLM-F which mainly comprised flavonoids (45.14%), phenolic acids (16.05%), organic acids (14.59%), terpenoids (7.57%), and alkaloids (5.10%), among which histidinol was predominant at 4.60%, followed by quercetin 7-O-β-D-glucuronide (3.79%), malonic acid (2.78%), tubuloside C (2.69%) and 3-hydroxybutyrate (2.57%). In total, 467 compounds were identified in RDP-F which mainly included flavonoids (41.26%), phenolic acids (16.06%), organic acids (13.32%), terpenoids (13.21%), and alkaloids (3.96%), and the top five predominated metabolites were camaldulenic acid (2.99%), poncirin (2.80%), luteolin-caffeoyl-O-rhamnoside (2.77%), tiliroside (2.77%), and histidinol (2.45%). There were 423 compounds detected from RSL-F which mainly comprised flavonoids (35.13%), terpenoids (17.18%), organic acids (16.33%), phenolic acids (12.35%), and alkaloids (5.10%), in which histidinol (4.09%) had the largest proportion of content, followed by hederagenin (2.45%), alphitolic acid (2.44%), pomolic acid (2.43%), and maslinic acid (2.43%). Detailed information is provided in Appendix A. In summary, there were significant differences in the main component types and metabolites with the highest content among the five species, especially the triterpenoids, with the highest content detected in RSS-F, which is different from the other four with the highest content being flavonoids. Histidinol was a common component among the five samples with the highest content, whereas the other main components varied.

### 3.2. Multivariate Analysis

#### 3.2.1. PCA and Hierarchical Clustering Analysis

PCA projects data in a reduced hyperspace defined by principal components (PCs), which are linear combinations of the original variables, with the first component exhibiting the largest variance. PCA facilitates data exploration in multivariate datasets, demonstrating the pattern of similarity among observations by distributing them as points on maps (score plots) [36]. The PCA score plot showed that all samples were clearly separated into five distinct clusters, consistent with the five fruits of *Rosa* spp. (Figure 2A,B). The first and second PCs (PC1 and PC2, respectively) accounted for 30.62% and 26.84% of the total variance, respectively. In addition, the mixed samples clustered and were located near the center, revealing the systematic stability and repeatability of the analytical method.

The results of the hierarchical clustering analysis better responded to the metabolite characteristics in the fruit samples of different *Rosa* resources (Figure 3). According to the metabolites cluster analysis of each resource, the five *Rosa* samples were divided into the following two classes: the first class was RLM-F and RSL-F; and the second class was RDP-F, RSS-F, and RRT-F. The second class could further be divided into two subclasses, and the first subclass was RDP-F, RSS-F, and RRT-F, which indicated that the samples contained in each category had similar metabolites. The results also showed the distance between the genetic relationships among different resources.

#### 3.2.2. OPLS-DA

OPLS-DA is a supervised model that reduces system noise, extracts the components of the independent variable X and dependent variable Y, and calculates the correlation between these components [37]. Compared with PCA, OPLS-DA can maximize inter-group differentiation, thus facilitating the search for different metabolites. The difference between groups is seen on the horizontal axis, which represents the characterized PC, whereas the difference within the group is seen on the vertical axis, which represents orthogonal PCs [38]. In this study, pretreated (par-scaling method) OPLS-DA models were used to further explore *Rosa* extracts to maximize the covariance between RRT-F and the other four *Rosa* groups. Three OPLS-DA score plots of the tested samples from the five species showed excellent accuracy, with R^2^Y (cum) and Q^2^ (cum) values greater than 0.5 [39]. Specifically, the R^2^Y (cum) and Q^2^ (cum) values were 1 and 0.997, respectively, in the extracts of RRT-F and RSS-F (Figure 4D); 1 and 0.999, respectively, in the extracts of RRT-F and RLM-F (Figure 4F); 1 and 0.998, respectively, in the extracts of RRT-F and RDP-F (Figure 4H); and 1 and 0.999, respectively, in the extracts of RRT-F and RSL-F (Figure 4H), indicating a significant difference between RRT-F and the other four *Rosa* fruits.

### 3.3. Differentially Expressed Metabolite Analysis of the Five Rosa Fruits

OPLS-DA and one-way analysis of variance were performed to determine significant differences between metabolites among the five *Rosa* samples, of which log [FC] ≥ 2 or ≤0.5, VIP ≥ 1, and *p* < 0.05 indicated significant differences between the metabolites. In total, 486 significantly different metabolites were identified (Appendix A). The relative percentages of differential metabolites in RRT-F, RSS-F, RLM-F, RDP-F, and RSL-F were 88.08, 84.71, 84.44, 91.25, and 87.78%, respectively, including chemicals such as flavonoids (22.29–45.13%), phenolic acids (17–22.27%), terpenoids (7.7–24%), and organic acids (2.29–8.56%). In addition, relatively low levels of lignans (0.59–4.06%), alkaloids (3.17–8.02%), vitamins (0.68–0.98%), and coumarins (0.09–0.5%) were detected (Table 1). In relation to RRT-F, the upregulated and downregulated metabolites were 207 and 65, respectively, in RSS-F; 219 and 103, respectively, in RLM-F; 96 and 188, respectively, in RDP-F; and 212 and 116, respectively, in RSL-F (Figure 5A). Among the differential metabolites, the top three compounds with the highest content were flavonoids, phenolic acids, and triterpenes. These compounds are also the most commonly reported components of *Rosa* species.

#### 3.3.1. Differentially Expressed Flavonoids in *Rosa* Samples

Flavonoids are natural biological response modifiers of plant polyphenolic compounds which have been linked to many pharmacological effects, such as anti-inflammatory, antioxidant, anti-allergic, anti-microbial, and anticancer activities [40,41,42,43]. Flavonoids are the second most abundant constituent in both *R. roxburghii* and *R. sterilis* fruits [6]. As shown in Appendix A, 175 differential flavonoid compounds, including the subclasses of 76 flavonols, 44 flavonoids, 18 flavanols, 13 flavanones, 10 isoflavones, six anthocyanins, five proanthocyanidins, and three chalcones, were detected in the five samples, according to the combination of FC ≥ 2 and *p* < 0.05. In relation to RRT-F, the upregulated and downregulated flavonoids were as follows: 83 and 29, respectively, in RSS-F; 66 and 46, respectively, in RLM-F; 27 and 74, respectively, in RDP-F; and 87 and 45, respectively, in RSL-F (Figure 5B). Figure 6 shows the relatively high flavonoid content of the five *Rosa* fruits. More comparative information on the differential metabolites is presented in Appendix A.

Among these flavonoids, some were more significantly abundant in the other four samples than those in RRT-F (Appendix A). Fourteen flavonoids were not found in RRT-F but were detected in the other four, one, or two samples. These components also exhibit unique biological activities or contribute to the formation of color or flavor in the species. For instance, except for RRT-F, the other four *Rosa* fruits contained quercetin-3-O-sambubioside, with RSL-F containing the highest relative content, which reached 160 times that of RSS-F and approximately 15–20 times that of RDP-F and RLM-F. Silibinin was only detected in RLM-F, RDP-F, and RSL-F, with a content as high as 2% in RLM-F, which is 14.2 times the content in RDP-F and 245.8 times the content in RSL-F. Myricetin (MYR) was detected only in RLM-F and RDP-F, with relative contents of 0.38 ± 0.02% and 0.02 ± 0.00%, respectively. Some typical flavonoids such as poncirin, luteolin-caffeoyl-O-rhamnoside, kaempferol 3-O-β-d-(6″-O-(E)-p-coumaroyl) glucopyranoside, catechin, and tiliroside have a higher content in RDP-F. Biorobin, rutin, quercetin-3-O-robinobioside, nicotiflorin, and quercetin derivatives were relatively high in RLM-F; quercetin 4′-O-glucoside was abundant in RSL-F (Appendix A).

#### 3.3.2. Differentially Expressed Phenolic Acids in *Rosa* Samples

Phenolic acids are well known for their immense dietary health benefits and functionalities, such as antioxidant, anti-inflammatory, immunoregulatory, anti-allergeric, anti-atherogenic, anti-microbial, anti-thrombotic, cardioprotective, and anti-cancer activities, along with antidiabetic properties [44,45]. As shown in Appendix A, 131 differential phenolic acids, including 66 hydroxybenzoic acids and 54 hydroxycyclic acids, were classified from the five samples using OPLS-DA. In relation to RRT-F, the following phenolic acids were upregulated and downregulated: 65 and 11, respectively, in RSS-F; 65 and 30, respectively, in RLM-F; 28 and 49, respectively, in RDP-F; and 63 and 29, respectively, in RSL-F (Figure 5C). Compared to RRT-F, RSS-F, RLM-F, RDP-F, and RSL-F demonstrated higher contents of gallic acid derivatives as the predominant phenolic acids (Appendix A). For instance, 1,4-di-O-galloyl-β-D-glucose and 1,6-Di-O-galloyl-D-glucose were not detected in RRT-F; however, both were distributed in the other four fruits, with RLM-F having the highest relative contents (1.52 ± 0.04% and 1.54 ± 0.12%, respectively). Methyl 6-O-galloyl-β-D-glucopyranoside was only detected in RSS-F and RSL-F, with the relative contents of 0.0346 ± 0.0199% and 0.1194 ± 0.0595%, respectively. The contents of 1-O-galloyl-β-D-glucose and 2-O-galloyl-β-D-glucose in RRT-F was significantly lower than those in the other four samples, with RLM-F exhibiting the highest content, approximately three times that of RRT-F (Appendix A). Gallic acid derivatives are intermediate metabolites which synthesize a series of gallitannins and ellagitanins and play an important role in plant metabolism [46]. Figure 7 shows the relatively high phenolic acid content of the five *Rosa* fruits. These results indicate the potential research value of RLM-F, RDP-F, and RSL-F. Heat maps for the other compounds are displayed in Appendix A.

#### 3.3.3. Differentially Expressed Terpenoids in *Rosa* Samples

As a major source of medicinal natural products, monoterpenes and sesquiterpenes of the genus *Rosa* mainly contribute to the aroma, and triterpenoids are one of its main functional components [47,48,49,50]. Among the 38 detected terpenoids, 37 exhibited significant differences among five *Rosa* samples which included four monoterpenes, two diterpenes, two sesquiterpenes and 29 triterpenoids (Appendix A). Compared to RRT-F, the following triterpenoids were upregulated and downregulated: 1 and 5 in RSS-F, respectively; 19 and 1 in RLM-F, respectively; 8 and 12 in RDP-F, respectively; and 5 and 9 in RSL-F, respectively (Figure 5D). Appendix A shows the terpenoids with significant differences in the other four fruits compared to RRT-F. For instance, the content of 3*β*-hydroxy-28-norurs-17,19,21-trien, terminolic acid, pomolic acid, and isoceanothic acid in RSS-F were all more than twice the amount as those in RRT-F. Geniposide, 2*α*-hydroxyursolic acid, and ursolic acid in RDP-F and methoxyursolic acid in RSL-F reached more than 20 times the content of those in RRT-F, respectively. These results indicate that in addition to RRT-F and RSS-F, the potential research value of RDP-F and RSL-F can also be taken seriously. Figure 8 shows the comparison of relatively high levels of terpenoids in five *Rosa* fruits.

#### 3.3.4. Potential Markers in Five Samples

Based on the relative contents in the heat map, we screened 125 compounds that could serve as potential markers for the five *Rosa* fruits (Appendix A), among which the 29 potential markers screened in RRT-F included nine flavonoids, 14 phenolic acids, and one terpenoid. The top five markers with the highest content were citraconic acid (2.38%), 7S,8R-threo-3′,9,9′-trihydroxy-3-methoxy-4′,7-epoxy-neolignan-4-O-α-L-rhamnopyranoside (1.61%), brevifolin carboxylic acid (0.50%), di-O-glucose-quinic acid (0.48%), and mirtillin (0.19%), respectively. In RSS-F, 10 compounds were screened as potential markers, including two flavonoids and three phenolic acids, among which O-phosphocholine was predominant (0.33%), followed by jasmonic acid (0.17%), cinnamic acid (0.13%), sinapaldehyde glucoside (0.07%), and 1-O-feruloyl quinic acid (0.04%). In RLM-F, 37 compounds were identified as potential markers, including 22 flavonoids, six phenolic acids, and three terpenoids. Tubuloside C (2.69%), nicotiflorin (1.86%), biorobin (1.77%), 3-O-digalloyl quinic acid (0.94%), and kaempferol 3-O-β-D-neohesperidoside (0.91%) were the top five compounds with the highest relative content in RLM-F. For RDP-F, 30 potential markers, including 11 flavonoids, nine phenolic acids, and three terpenoids were screened. The top five compounds with the highest relative content in RDP-F were 2α-hydroxyursolic acid (1.16%), ethyl gallate (1.13%), dunalianoside C (0.45%), L-(+)-tartaric acid (0.43%), and dihydrodehydrodiconiferyl alcohol 4-O-β-D-glucopyranosides (0.27%). Moreover, 19 potential markers were screened in RSL-F, including seven flavonoids, five phenolic acids, and two terpenoids. The top five compounds were anchoic acid (2.17%), spiraeoside (1.77%), myricetin-O-rhamnoside (1.36%), methoxyursolic acid (0.92%), and formononetin (0.58%). The structures of the top five potential markers with the highest content among the five fruit samples are shown in Figure 9.

### 3.4. Analysis of Differential Metabolic Pathways in the Five Rosa Fruits

To further explore differences in metabolic pathways among the five *Rosa* fruits, functional annotations were performed on the significantly different metabolites and the metabolic pathway enrichment of different metabolites using the KEGG databases. Based on the number of upregulated and downregulated metabolites, we plotted the metabolic pathway differential abundance scores of RSS-F, RLM-F, RDP-F, and RSL-F, relative to RRT-F.

There were 120 metabolites annotating five pathways between RSS-F and RRT-F. The KEGG pathway types with the highest number of annotated metabolites were metabolic pathways, the biosynthesis of secondary metabolites, flavonoid biosynthesis, flavone and flavonol biosynthesis, and phenylpropanoid biosynthesis pathways which accounted for 67.21, 54.10, 21.31, 18.03, and 13.11%, respectively. In addition to the metabolic pathways and biosynthesis of secondary metabolites, flavone and flavonol biosynthesis demonstrated the highest richness factors (Figure 10A). The rich factor was the ratio of the number of differentially expressed genes to the total number of annotated genes in a given pathway. The larger the rich factor, the greater the degree of enrichment [51]. As shown in the differential abundance score map (Figure 10E), the porphyrin and chlorophyll metabolism and alpha-linolenic acid metabolism pathways were significantly upregulated in RSS-F, compared with those in RRT-F. However, the enrichment level was relatively low, owing to fewer annotated metabolites.

A total of 124 metabolites were annotated in 54 pathways between RLM-F and RRT-F. The metabolic pathways with the highest number of annotated metabolites were the biosyntheses of secondary metabolites, flavonoids, phenylpropanoid, flavone, and flavonol (Figure 10B,F), accounting for 64.62, 55.38, 16.92, 13.85, and 12.31%, respectively. Zeatin biosynthesis, riboflavin metabolism, glycolysis/gluconeogenesis, and flavone and flavonol biosynthesis metabolic pathways were upregulated in RLM-F, compared with those in RRT-F. However, these metabolic pathways, particularly the flavone and flavonol biosynthetic pathways, had low enrichment levels.

In total, 124 metabolites were annotated in 58 pathways between RRT-F and RDP-F. The Biosyntheses of secondary metabolites, flavonoids, flavone and flavonol, phenylpropanoid, and isoflavonoid accounted for the top five metabolic pathways, with proportions of 63.01, 53.42, 26.03, 9.59, and 9.59%, respectively. Compared with RRT-F, RDP-F demonstrated the upregulation of nine metabolic pathways, with flavonoid and isoflavonoid biosynthesis demonstrating relatively higher rich factors. In contrast, tyrosine metabolism, the pentose phosphate pathway, carbon fixation in photosynthetic organisms, and 2-oxocarboxylic acid metabolism pathways had relatively fewer rich factors (Figure 10C,G).

A total of 122 metabolites were annotated in 48 pathways between the RRT-F and RSL-F groups. The pathways with the highest number of annotated metabolites were the metabolic pathways and the biosyntheses of secondary metabolites, flavonoids, phenylpropanoid, and flavone and flavonols, with proportions of 63.08, 53.85, 21.54, 12.31, and 10.77%, respectively. Information on the other metabolic pathways is shown in Figure 10I,J. Nine metabolic pathways were significantly upregulated in the RSL-F group, compared to those in the RRT-F group. In contrast, the biosynthesis pathways for ubiquinone and others like terpenoid-quinone, phenylalanine, tyrosine, and tryptophan, flavonoids, and anthocyanins, along with the alpha-linolenic acid metabolism pathway, were significantly downregulated in RSL-F. The biosyntheses of phenylpropanoids and secondary metabolites showed relatively higher rich factors (Figure 10D,H).

### 3.5. TPS Contents Detection

The phenol–sulfuric acid method was used to detect TPS content in the five *Rosa* fruits. The polysaccharide contents of the different *Rosa* fruits were found to be significantly different (Table 2). Among all the samples, RLM-F showed the highest TPS content/100 g (64.48 g), followed by those in RSL-F (53.42 g), RDP-F (49.11 g), RSS-F (39.06 g), and RRT-F (33.80 g).

### 3.6. Antioxidant Activity

#### 3.6.1. Cell Viability

The CCK-8 assay was performed to determine the effects of the five fruit extracts on HaCaT keratinocyte viability at three different concentrations [52]. The results in Figure 11A show that, at concentrations < 0.25 mg/mL, the viabilities of RRT-F and RSL-F cells were not significantly different from that of the control. In contrast, 0.2 mg/mL of RSS-F, RLM-F, and RDP-F exhibited low cytotoxicity. Therefore, 0.25 mg/mL of RRT-F and RSL-F and 0.2 mg/mL of RSS-F, RLM-F, and RDP-F extracts were used in all subsequent experiments.

#### 3.6.2. Effects of the Five *Rosa* Samples on H_2_O_2_-Induced Cytotoxicity in HaCaT Cells

The CCK-8 assay was performed to determine the protective effects of *Rosa* extracts on HaCaT cell viability under H_2_O_2_ damage. As shown in Figure 11B, the treatment of HaCaT cells with H_2_O_2_ significantly reduced cell viability to 55.44% relative to the control cells. However, the viability of the HaCaT cells treated with samples of 0.25 mg/mL RRT-F and RSL-F and 0.2 mg/mL RSS-F, RLM-F, and RDP-F prior to H_2_O_2_ exposure significantly increased to 71.58, 69.25, 61.68, and 69.55%, respectively. RSL-F alone did not exhibit significant cell-damage-repairing effects.

#### 3.6.3. Effects of the Five *Rosa* Samples on SOD and GSH-Px Activities and MDA Content in H_2_O_2_-Induced HaCaT Cells

To explore the protective effects of the five *Rosa* samples on antioxidant defense systems, the activities of SOD and GSH-Px and the MDA content were investigated using the respective kits. Compared to the controls, HaCaT cells treated with H_2_O_2_ alone showed significantly decreased activities of SOD and GSH-Px, as well as increased MDA content (*p* < 0.05). After treatment with 0.25 mg/mL of RRT-F and RSL-F and 0.2 mg/mL of RSS-F and RDP-F, SOD activity significantly increased in the H_2_O_2_-induced HaCaT cells, compared to that in the H_2_O_2_-treated group (Figure 11C). Similarly, treatment with 0.25 of mg/mL RRT-F and RSL-F, and 0.2 mg/mL of RSS-F, RLM-F, and RDP-F significantly increased GSH-Px activity (Figure 11D) and significantly decreased the MDA content (Figure 11E) in H_2_O_2_-induced HaCaT cells, compared with those in H_2_O_2_-treated cells. RRT-F exhibited the highest SOD and GSH-Px activity, whereas RLM-F showed the highest inhibitory effect on the H_2_O_2_-induced increase in MDA content. Excess reactive oxygen species (ROS) can oxidize lipids in vivo, eventually forming MDA, which can direct the cross-linking of proteins and nucleic acids, thus exhibiting cytotoxicity. Therefore, the MDA content may directly reflect the level of lipid peroxidation in the body and indirectly reflect the level of ROS-induced cell damage. In summary, all five *Rosa* fruit samples exhibited significant antioxidant activity. However, antioxidant mechanisms varied among different *Rosa* species.

### 3.7. Pearson Correlation Analysis

To further elucidate the relationship between the major markers and antioxidant activity, Pearson’s correlation analysis was performed among SOD and GSH-Px activities, the MDA content, and 88 markers (Figure 12). According to Pearson correlation analysis, SOD and GSH activities showed a high correlation. Therefore, we classified the metabolites related to the two and identified 21 metabolites demonstrating a high correlation with both SOD and GSH-Px activity. MYR restores H_2_O_2_-induced cellular antioxidant activity against SOD and GSH-Px [53]. Among the five *Rosa* fruits, RLM-F had the highest MYR content, followed by that in RDP-F, which significantly exceeded that of the other three samples. In addition, 29 metabolites, including 23 flavonoids and six phenolic acids, showed significant negative correlations with the MDA content. Interestingly, all 21 compounds that positively correlated with SOD and GSH-Px activities were higher in RRT-F than those in the other four *Rosa* fruits. Nevertheless, the 29 compounds negatively correlating with the MDA content were all found to have the highest content in RLM-F, remarkably exceeding that in the other four fruits. Overall, all five *Rosa* fruits exhibited significant antioxidant activities; however, their antioxidant mechanisms differed.

## 4. Discussion

Throughout history, many fruits domesticated by humans come from the *Rosaceae* family. The pseudo-fruits of different *Rosa* species, which are called rose hips, contain high amounts of vitamin C and other beneficial biological active compounds such as phenolics, carotenoids, carbohydrates, and fatty acids [54]. Rose hips can be consumed fresh or used in food products such as herbal tea, jam, jelly, syrup, or wine. These days, rose hips are used as a component in probiotic products [55,56]. However, so far, a large amount of research has only focused on a few species, especially *Rosa rugosa* which is mainly used for ornamental and edible purposes, as well as *R. roxburghii* which is mainly used for food and medicine. There are still a lot of fruit resources of the genus *Rosa* worth developing which have not received sufficient attention from researchers. Most studies on the chemistry of RRT-F have focused on polysaccharides, which were hypothesized to be the basic components responsible for the immune-related effects of RRT-F [57]. In our study, the TPS of five *Rosa* fruits were determined and compared, and the TPS content of the other four rose hips was significantly higher than that of RRT-F, especially the TPS content of RLM-F which was almost twice that of RRT-F. This is the first time that the polysaccharide content of five *Rosa* fruits has been compared under the same conditions. The results of this study also demonstrate that, except for RRT-F, there is a higher polysaccharide content in the other four rose hips, especially in the RLM-F fruit. In fact, the polysaccharides in RLM-F have also been proven in many studies to have immunomodulatory effects [58]. Therefore, it is speculated these four fruits have considerable immune-enhancing effects. Encouragingly, the determination of polysaccharide content has been implemented for quality control of RLM-F [57]. Therefore, further in-depth and systematic research is needed on the isolation and mechanism of polysaccharides from *Rosa* plants.

Although previous studies have found that the types of compounds in *Rosa* species have certain similarities (for example, most researched *Rosa* species all contain large amounts of phenolic acids, flavonoids, triterpenes, and tannins components), this study proved that different species of *Rosa* plants have significant differences in the distribution and content of different component types, while the differences in component distribution and content among different species may be the main reason for their differences in taste and medicinal efficacy.

Based on the fact that the compositions and efficacy of RRT-F are the most comprehensively researched among the selected species, in order to more reasonably evaluate the potential value of these rose hips, we mainly used RRT-F as a reference and made a comprehensive comparison between the components of the other four rose hip fruits and RRT-F. Previous research has no comprehensive comparison between these five edible fruits. The results showed that there are significant differences in the compositions of the four fruits compared to RRT-F. We subsequently conducted a detailed comparative analysis on the flavonoids, phenolic acids, and terpenoids with significant differences.

Liu initially compared the multiple constituents of RRT-F and RSS-F using the UFLC/Q-TOF-MS/MS method which proved that the chemical profiles of RRT-F and RSS-F showed obviously differences based on the representative negative signals. They compared the difference in organic acid and flavonoids between the two, leading to the result that 9,12,15-octadecatrienoic acid was only detectable in RSS-F, whereas syringic acid was only found in RRT-F. From the perspective of flavonoids, epigallocatechin (EGC) was only found in RSS-F rather than in RRT-F [6]. However, in our present study, EGC was detected in both RRT-F and RSS-F. Among the five fruits, except for RSL-F, which was not detected, there was not much difference in EGC content among the other four species. In addition, syringic acid was detected in all five *Rosa* fruits. This may be closely related to differences in collection time or habitat. Compared to RRT-F, a lot of flavonoid components were only detected in the other four fruits, which contribute to their functional qualities. For instance, quercetin-3-O-sambubioside was only found in the other four fruits rather than in RRT-F, of which, RSL-F contained the highest relative content. Quercetin-3-O-sambubioside possesses antiproliferative [59] and antioxidant [60] activities and also can effectively promote excitation at the neural center [61]. Silibinin can suppress T-cell-dependent liver injury as an immune response modifier [62]. In addition, silibinin is a potential anticancer [63,64] and therapeutic agent for treating Alzheimer’s disease [65]. Myricetin (MYR) was only detected in RLM-F and RDP-F, with relative contents approximately of 0.3815% and 0.0252%, respectively. MYR is recognized mainly for its iron-chelating, antioxidant, anti-inflammatory, anticancer, antibacterial, antiviral, and anti-obesity effects; cardiovascular protection; protection against neurological damage; and protection of the liver against potential injuries [66,67]. European countries have developed and marketed health products containing MYR, owing to its antioxidant and cholesterol-lowering effects [66]. In this study, the most detected potential markers in the five fruits were phenolic acids and flavonoids. This result further confirmed that there were significant differences in the phenolic acid and flavonoid components of the five fruits.

Apart from the above two components, triterpenoids were the other main constituents in *Rosa* species which have been reported to have various health functions such as anti-inflammatory [68,69], anti-acetylcholinesterase, neuroprotective [68], antifungal [49], and immunomodulatory activity [48]. Currently, there are three types of triterpenoids and their derivatives found in *Rosa* species: ursolic acid, oleanolic acid, and lupinic acid [10]. Among the differential metabolites detected in this study, there were 29 triterpenoid compounds. Among them, corosolic acid was the triterpenoid with the highest content in RLM-F as opposed to RRT-F. Accumulating evidence has indicated that corosolic acid exerts anti-diabetic, anti-obesity, anti-inflammatory, anti-hyperlipidemic and anti-viral effects. More importantly, corosolic acid has recently attracted much attention due to its anticancer properties and innocuous effects on normal cells [50,70]. Moreover, 3β-hydroxy-28-norurs-17,19,21-trien, terminolic acid, and isoceanothic acid were first reported in RSS-F and with more than twice the abundance than those in RRT-F. Otherwise, 2α-hydroxyursolic acid and ursolic acid in RDP-F and methoxyursolic acid in RSL-F reached more than 20 times the content of those in RRT-F, respectively. Ursolic acid and its derivatives have various pharmacological effects, including anti-inflammatory, hepatoprotective, antitumor, cardioprotective, neuroprotective, antimicrobial, antihyperlipidemic, anti-diabetic, antifungal, antiviral and trypanocidal effects, which have been used for a long time in folk medicine [71,72,73]. Overall, there is still relatively little attention being paid to the triterpenoid components in the fruits of the *Rosa* genus. The triterpenoid components of these five *Rosa* fruits have high potential for application and deserve in-depth research on their isolation and mechanisms.

Due to the abundance of typical antioxidant components such as phenolic acids and flavonoids in *Rosa* species, antioxidant activity has always been an important indicator for evaluating their potential applications. Usually, these medicinal and edible plants are increasingly favored by the food and cosmetics industries due to their safety and effectiveness. The results of this study strongly confirmed the antioxidant activity of the five *Rosa* fruits. Among them, RRT-F showed the best cell damage repair ability in the HaCaT cell model damaged by H_2_O_2_, possibly due to its extremely high levels of ascorbic acid and SOD. Similarly, RRT-F also exhibited the highest SOD- and GSH-Px-increasing activity. However, we surprisingly found that RLM-F showed the highest inhibitory effect on the H_2_O_2_-induced increase in MDA content. We speculated it may be related to the highest content of polysaccharides in RLM-F, to a certain extent, while the specific mechanism needs further research. The results of Pearson correlation analysis further confirmed that flavonoids and phenolic acids could serve as the main types of antioxidant components in the five *Rosa* species.

## 5. Conclusions

In summary, this study used *R. roxburghii* fruit as a control to comprehensively compare the differences in the composition and quality of several *Rosa* fruits and evaluated their antioxidant activities using metabolomic methods and HaCaT cell models. A total of 514 metabolites were identified via the UPLC-ESI-MS/MS approach, predominated by flavonoids and phenolic acids, which accounted for 33.66% and 26.67% of the total metabolites, respectively. Among the differential metabolites, the three categories with the highest quantity and content were flavonoids, phenolic acids, and terpenoids. A total of 125 compounds could serve as potential markers for the five *Rosa* fruits. Among which, 29 potential markers were screened in RRT-F. In RSS-F, 10 compounds were screened as potential markers. In RLM-F, 37 compounds were identified as potential markers. For RDP-F, 30 potential markers were detected, and 19 potential markers were screened in RSL-F. KEGG database analysis revealed that the differential metabolites in the fruits of the five *Rosa* species were mainly distributed across the following five metabolic pathways: metabolic pathways and the biosynthesis of secondary metabolites, flavonoids, phenylpropanoids, and flavone and flavonols. All five *Rosa* extracts effectively decreased MDA levels while increasing the activities of SOD and GSH-Px in the H_2_O_2_-induced HaCaT cell model, demonstrating a high potential for antioxidant development. In summary, this study provides comprehensive insights into the underlying metabolic causes of taste variation and antioxidant activity in various *Rosa* fruits and provides additional resources for the discovery of dietary supplements. These results will also be helpful for understanding the differences in the metabolites of different *Rosa* fruits and for evaluating their potential for further development and utilization in foods and cosmetics.

## Figures and Tables

**Figure 1 foods-13-00796-f001:**
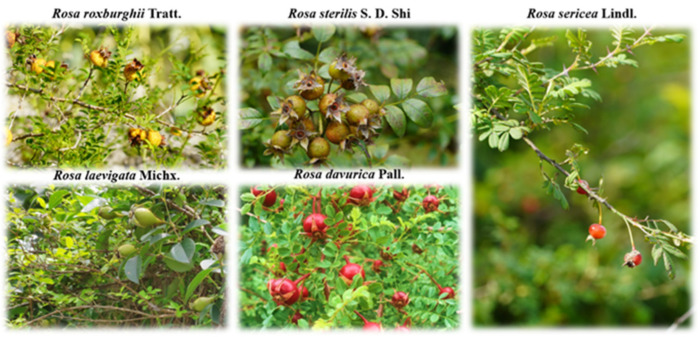
Morphology of five rose hips.

**Figure 2 foods-13-00796-f002:**
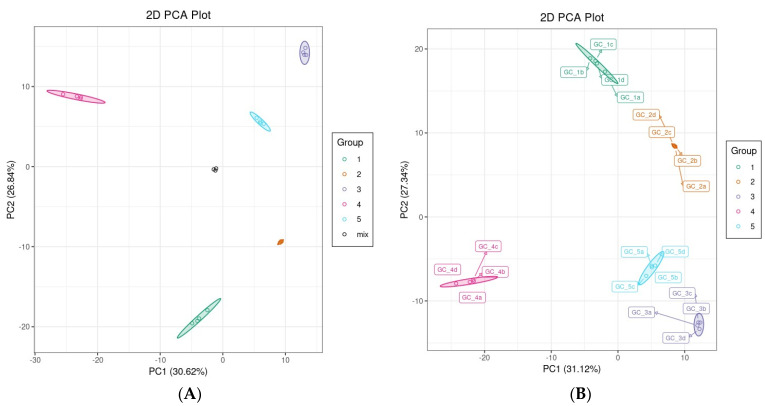
(**A**) Score plot of principle component analysis (PCA) in five samples with quality control sample, and (**B**) score plot of PCA without quality control sample. GC_1, RRT-F; GC_2, RSS-F; GC_3, RLM-F; GC_4, RDP-F; GC_5, RSL-F.

**Figure 3 foods-13-00796-f003:**
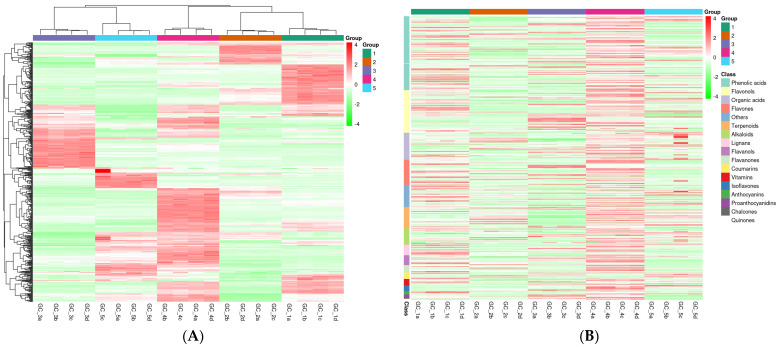
(**A**) The HCA heat map analysis of *Rosa* samples with clustering and (**B**) with compound classifications. The color indicates the level of accumulation of each metabolite, from low (green) to high (red). GC_1, RRT-F; GC_2, RSS-F; GC_3, RLM-F; GC_4, RDP-F; GC_5, RSL-F.

**Figure 4 foods-13-00796-f004:**
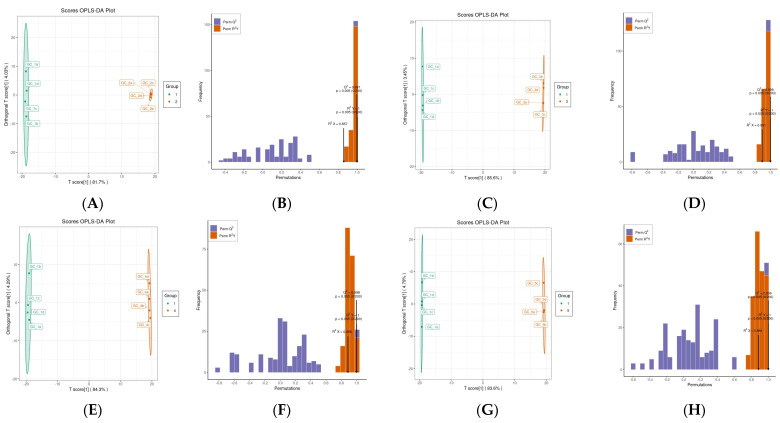
(**A**,**C**,**E**,**G**) Score plot generated from orthogonal partial least squares discriminant analysis (OPLS-DA), and (**B**,**D**,**F**,**H**) OPLS-DA models of five samples. (**A**,**B**) RRT-F vs. RSS-F; (**C**,**D**) RRT-F vs. RLM-F; (**E**,**F**) RRT-F vs. RDP-F; (**G**,**H**) RRT-F vs. RSL-F.

**Figure 5 foods-13-00796-f005:**
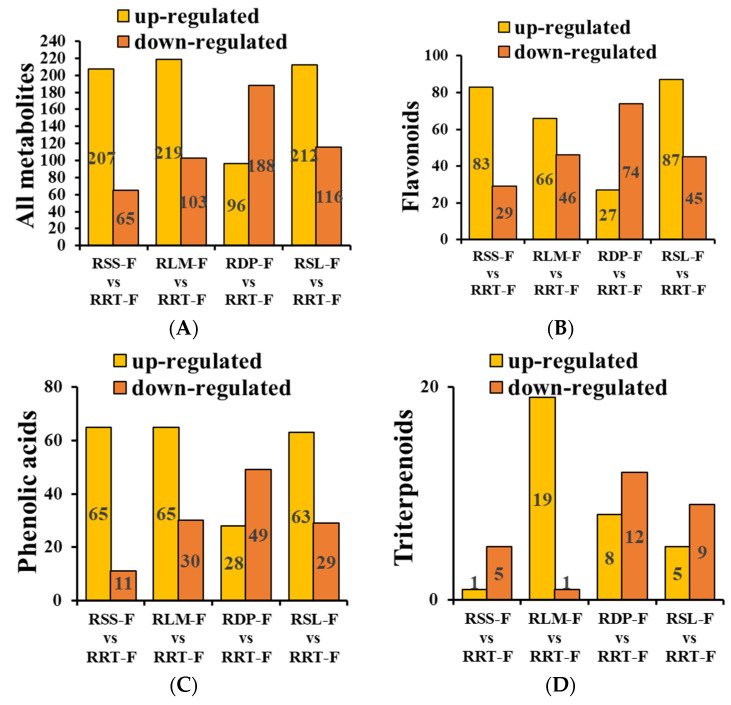
(**A**) The upregulated and downregulated metabolites from *Rosa sterilis* fruit (RSS-F), *Rosa laevigata* fruit (RLM-F), *Rosa davurica* fruit (RDP-F), and *Rosa sericea* fruit (RSL-F) compared with *Rosa roxburghii* fruit (RRT-F). (**B**) The upregulated and downregulated flavonoids, (**C**) phenolic acids, and (**D**) triterpenoids from RSS-F, RLM-F, RDP-F, RSL-F compared with RRT-F.

**Figure 6 foods-13-00796-f006:**
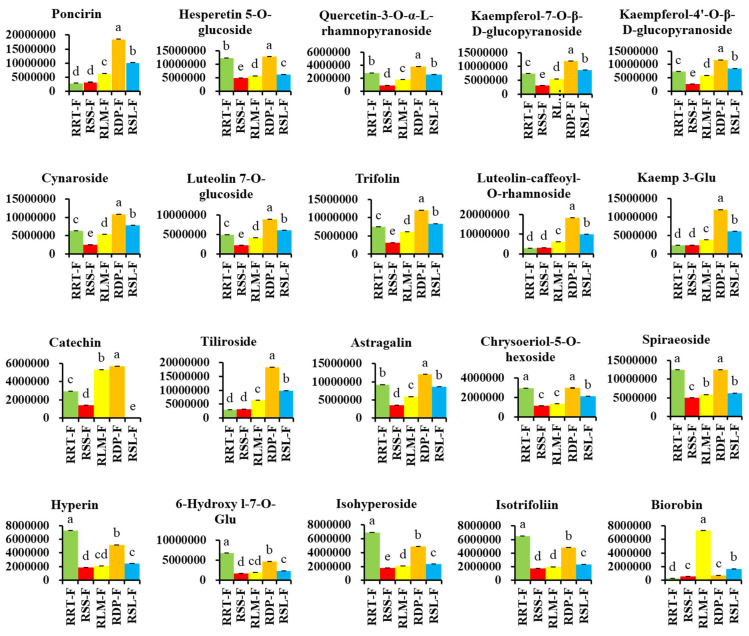
Comparing the relative contents of differentially expressed flavonoids with the highest total content in five rose hips based on their peak area. Kaemp 3-Glu, kaempferol 3-O-β-d-(6″-O-(E)-p-coumaroyl) glucopyranoside; 6-hydroxy l-7-O-Glu, 6-hydroxykaempferol-7-O-glucoside. Different letters on the bars represent significant differences between *Rosa* fruits samples (*p* < 0.05) analyzed via a post hoc Tukey’s test.

**Figure 7 foods-13-00796-f007:**
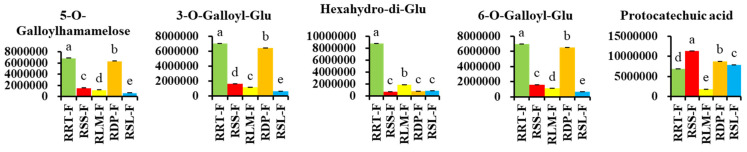
Comparing the relative contents of differentially expressed phenolic acids with the highest total content in five rose hips based on their peak area. 3-O-Galloyl-Glu, 3-O-Galloyl-β-D-glucose; 6-O-Galloyl-Glu, 6-O-Galloyl-β-D-glucose; 2-O-Galloyl-Glu, 2-O-Galloyl-β-D-glucose; 1-O-Galloyl-Glu, 1-O-Galloyl-β-D-glucose; 3,4,5-Glu, 3,4,5-trimethoxyphenyl-β-D-glucopyranoside; 5-Hydroxy-MF, 5-hydroxymethylfurfural. Different letters on the bars represent significant differences between *Rosa* fruits samples (*p* < 0.05) analyzed via a post hoc Tukey’s test.

**Figure 8 foods-13-00796-f008:**
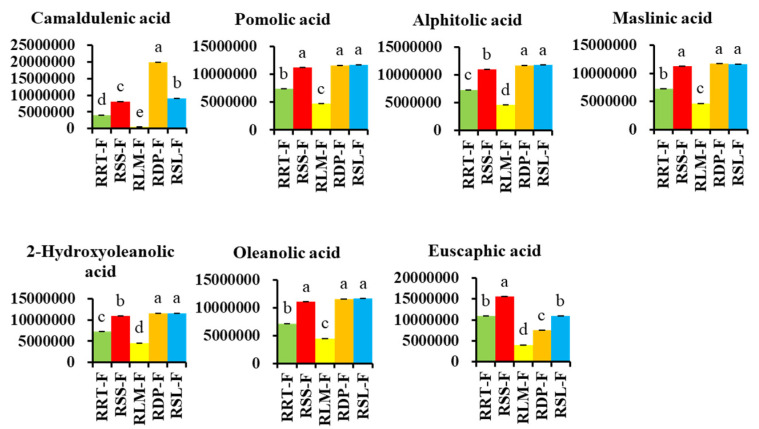
Comparing the relative contents of differentially expressed terpenoids with the highest total content in five rose hips based on their peak area. Different letters on the bars represent significant differences between *Rosa* fruits samples (*p* < 0.05) analyzed via the post hoc Tukey’s test.

**Figure 9 foods-13-00796-f009:**
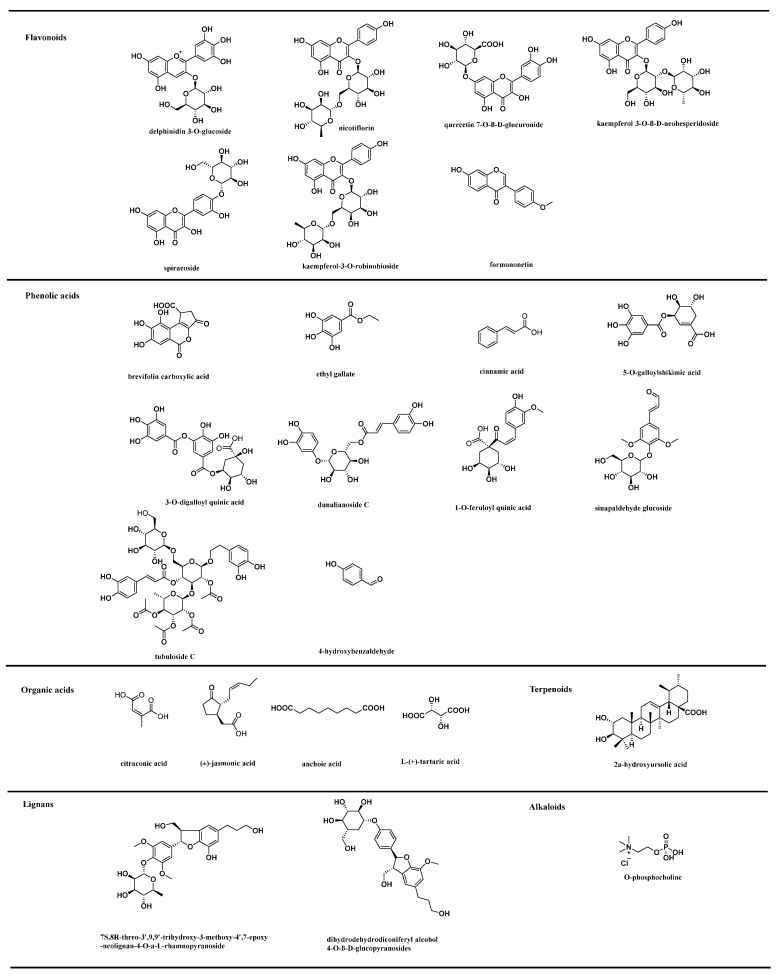
The top five potential markers with highest content in five samples.

**Figure 10 foods-13-00796-f010:**
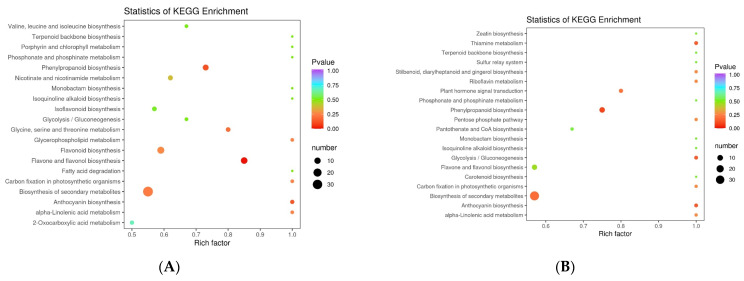
(**A**,**B**,**C**,**D**) KEGG enrichment map and (**E**,**F**,**G**,**H**) differential abundance score map of different metabolites in five samples. (**I**,**J**,**K**,**L**) KEGG bar plot of DEMs in five samples. (**A**,**E**,**I**) RSS-F vs. RRT-F; (**B**,**F**,**J**) RLM-F vs. RRT-F; (**C**,**G**,**K**) RDP-F vs. RRT-F; (**D**,**H**,**L**) RSL-F vs. RRT-F.

**Figure 11 foods-13-00796-f011:**
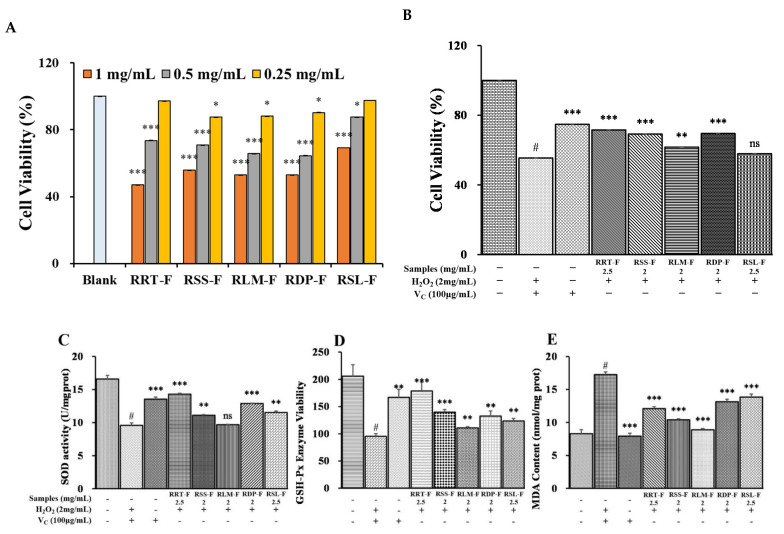
(**A**) Effects of samples on the cellular viabilities in HaCaT cells. (**B**) Recovery effect of samples on HaCaT cells damaged by H_2_O_2_. (**C**) Enhancing effects of *Rosa* samples on SOD viability and (**D**) GSH-Px enzyme viability in cellular lysates of HaCaT keratinocytes. (**E**) *Rosa* samples influenced the MDA levels in H_2_O_2_-damaged HaCaT cells. V_C_ (0.1 mg/mL) was used as a positive control. #, *p* < 0.05 versus the blank control. *, *p* < 0.05; **, *p* < 0.01; ***, *p* < 0.001 versus the non-treated control.

**Figure 12 foods-13-00796-f012:**
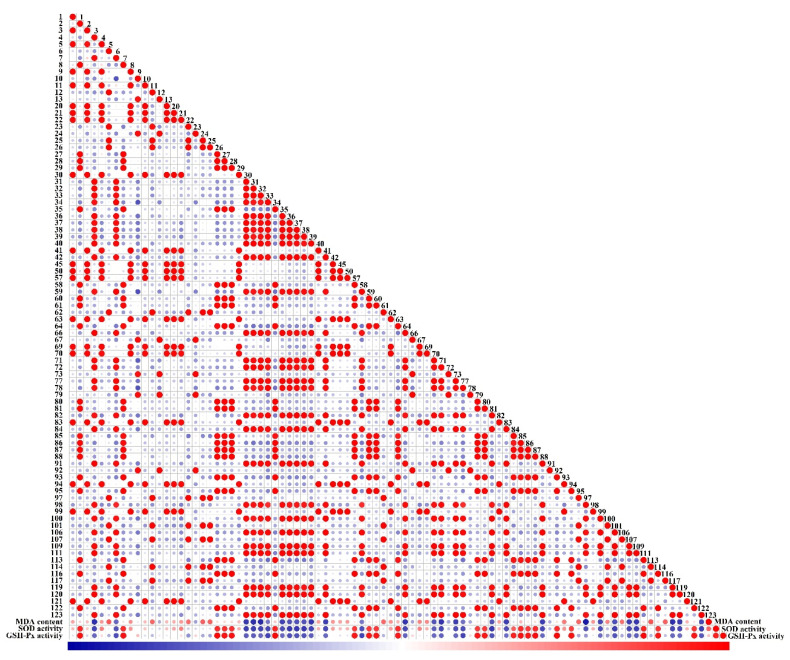
Correlation index between antioxidant activities and the potential markers (flavonoids and phenolic acids). The numbers represent the compounds in Appendix A. The red color indicates the positive correlation, the blue color indicates the negative correlation, and the blank represents no correlation. The size of the circle represents the significance of the corresponding correlation.

**Table 1 foods-13-00796-t001:** Relative proportion of peak area of potential DEMs in five *Rosa* samples.

Class	RRT-F	RSS-F	RLM-F	RDP-F	RSL-F
Flavonoids	33.98%	22.29%	45.13%	41.24%	35.08%
Phenolic acids	17.18%	12.67%	14.34%	15.20%	11.27%
Terpenoids	12.22%	24.81%	7.70%	16.21%	19.07%
Organic acids	12.29%	9.77%	8.56%	9.59%	12.18%
Lignins	4.06%	1.50%	1.29%	1.30%	0.59%
Alkaloids	3.45%	8.02%	3.40%	3.17%	3.77%
Vitamins	0.82%	0.87%	0.98%	0.68%	0.79%
Coumarins	0.25%	0.23%	0.44%	0.09%	0.53%
Others	3.84%	4.54%	2.60%	3.77%	4.52%
Total	88.08%	84.71%	84.44%	91.25%	87.78%

**Table 2 foods-13-00796-t002:** Contents of TPS of five *Rosa* samples.

Samples	Contents (g/100 g ± SD, n = 3)
	TPS
RRT-F	33.80 ± 0.07 ^c^
RSS-F	39.06 ± 0.04 ^bc^
RLM-F	64.48 ± 0.07 ^a^
RDP-F	49.11 ± 0.08 ^abc^
RSL-F	53.42 ± 0.10 ^ab^

Different letters for each parameter denote statistically significant differences between each other at *p* < 0.05.

## Data Availability

The original contributions presented in the study are included in the article/Appendix A, further inquiries can be directed to the corresponding authors.

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
