# Peer review of "UPLC-ESI-MS/MS-Based Analysis of Various Edible Rosa Fruits Concerning Secondary Metabolites and Evaluation of Their Antioxidant Activities"

_foods, 2024, doi:10.3390/foods13050796_

Round 1
Reviewer 1 Report
Comments and Suggestions for Authors
Manuscript ID foods-2883468
First of all, I didn’t find the supplementary materials. The link reported in the manuscript does not work.
Then, although the manuscript is interesting and a lot of work has been done, the authors are requested to rewrite it in a more orderly manner.
Abstract
Line 16: please, remove ‘common’
Introduction
Line 46: ‘is more than 5 billion Renminbi..’…I think something has been omitted. Please, correct since the sentence is not clear.
Between lines 51 and 52; please, remove the blanks
Please, could you add information on the diffusion or use of these fruits in other continents?
Materials and Methods
Line 99: please, remove the second ‘medium’
Line 114: please, specify the drying conditions since they can affect the antioxidant contents
Fig. 1B must be separated from Fig. 1A (it must become a new figure, i.e.. Fig. 2 because is part of the results and not of materials and methods). Obviously, Fig. 1A must become Fig. 1 and the numbers of all the other following figures must be changed. In addition, Fig. 1B indicates a generic composition. It is not clear to what of the cultivar it is referred. The same thing in the text comprised within lines 217-223.
Table 1 is not useful. Please, remove it.
Lines 122-123: please, remove the following sentence since it is a repetition “Briefly, the 122 five fruit samples were air-dried under laboratory conditions and ground into small pieces. “
Line 131: please, specify the filter material.
Line 162: it is not clear “The prediction parameters for the evaluation model are R2X, R2Y, and Q2.”
Line 168: what is TPS????
Line 184: what is the “enzyme-labeling instrument”?
Fig. 2 and following: authors must use only one acronym for each of the samples, otherwise the manuscript is confusing and difficult to read. It is not enough to produce a lot of data, you also need to report it in an orderly manner.
Line 280: ‘prediction’? This is a work to characterize, not to predict..It is not clear what authors mean.
Fig. 5a and throughout the text: what is the meaning of ‘up-regulated’ and ‘down-regulated’?
Author Response
Dear reviewers and editors:
Thank you for your kindly comments and suggestions regarding our manuscript. We have modified the manuscript accordingly, and the amendments were highlighted in yellow in the manuscript. The detailed corrections are listed below point by point:
Abstract
Line 16: please, remove ‘common’
Response: Thanks for your kindly comment, the redundant word has been deleted now.
Introduction
Line 46: ‘is more than 5 billion Renminbi..’…I think something has been omitted. Please, correct since the sentence is not clear.
Response: Thanks for your valuable comment, the sentence has been corrected accordingly.
Between lines 51 and 52; please, remove the blanks
Response: Thanks for your kindly comment, the blank has been removed now.
Please, could you add information on the diffusion or use of these fruits in other continents?
Response: Thanks for your kindly comment. The first three species are endemic to China and have only been studied in China. The latter two are distributed in Asian countries, and the distribution and application have been added now.
Materials and Methods
Line 99: please, remove the second ‘medium’
Response: Thanks for your kindly comment, the redundant word has been deleted now.
Line 114: please, specify the drying conditions since they can affect the antioxidant contents
Response: Thanks for your kindly comment, the drying conditions have been added now.
Fig. 1B must be separated from Fig. 1A (it must become a new figure, i.e.. Fig. 2 because is part of the results and not of materials and methods). Obviously, Fig. 1A must become Fig. 1 and the numbers of all the other following figures must be changed. In addition, Fig. 1B indicates a generic composition. It is not clear to what of the cultivar it is referred. The same thing in the text comprised within lines 217-223.
Response: Thanks for your kindly comment, Figure 1 has been adjusted as required. In addition, the statistical results of Figure 1B are meaningless and have been deleted. The statistical results of the five species metabolite classifications have been added to Table S5. and the analysis have been added to the text.
Table 1 is not useful. Please, remove it.
Response: Thanks for your kindly comment, Table 1 has been removed now.
Lines 122-123: please, remove the following sentence since it is a repetition “Briefly, the 122 five fruit samples were air-dried under laboratory conditions and ground into small pieces. “
Response: Thanks for your kindly comment, the sentence has been removed now.
Line 131: please, specify the filter material.
Response: Thanks for your kindly comment, the filter material has been specified now.
Line 162: it is not clear “The prediction parameters for the evaluation model are R2X, R2Y, and Q2.”
Response: Thanks for your kindly comment, the redundant sentence has been deleted now.
Line 168: what is TPS????
Response: Thanks for your kindly comment. The full name of the abbreviation” TPS” is now provided now.
Line 184: what is the “enzyme-labeling instrument”?
Response: Thanks for your kindly comment. The “enzyme-labeling instrument” here has been corrected to “microplate reader” now.
Fig. 2 and following: authors must use only one acronym for each of the samples, otherwise the manuscript is confusing and difficult to read. It is not enough to produce a lot of data, you also need to report it in an orderly manner.
Response: Thanks for your kindly comment. The samples acronym have been adjusted and additional annotations have been made in Figures.
Line 280: ‘prediction’? This is a work to characterize, not to predict..It is not clear what authors mean.
Response: Thanks for your kindly comment. The “predict” here is inappropriate, we have corrected now.
Fig. 5a and throughout the text: what is the meaning of ‘up-regulated’ and ‘down-regulated’?
Response: Thanks for your kindly comment. The ‘up-regulated’ used in this paper is to express that compared to RRT-F, metabolites in RSS-F, RLM-F, RDP-F, and RSL-F has a higher content; and ‘down-regulated’ means compared to RRT-F, metabolites in RSS-F, RLM-F, RDP-F, and RSL-F has a lower content. If there are any inappropriate aspects, please provide your feedback.
Thanks for your consideration, and please contact me if you have any questions.
Kind regards,
Wei Gu, PhD
E-mail: guwei2009@126.com
Reviewer 2 Report
Comments and Suggestions for Authors
Review on manuscript: foods-2883468
UPLC-ESI-MS/MS Based Analysis of Various Edible Rosa Fruits Concerning Secondary Metabolites and Evaluation of Their Antioxidant Activities
by Ming Ni, Junlei Chen, Mao Fu, Huanyang Li, Shengqian Bu, Xiaojiang Hao, Wei Gu
submitted to Foods
In the manuscript submitted for review, the authors studied secondary metabolites and antioxidant activities of various edible rosa fruits.
Detailed recommendation:
line 22 – such marking of the sample will not tell the reader anything,
line 22 – abbreviation should be explained upon first use,
line 114 – drying condition should be provided,
Figure 1B – are these data the authors' results? or literature data?
line 123 – drying condition should be provided,
line 126 – concentration temperature should be provided,
line 127 – freeze-drying condition should be provided,
line 131 – instead of rpm, the centrifugal force should be given,
lines 132-137 – website addresses are not needed here,
line 157 – typically, a description of the statistical methods used is provided at the end of the methodology section,
lines 168-180 – it is unclear for what purpose the carbohydrate content was determined, do they have any connection with the antioxidant and bioactive substances detected?
line 184 – model, manufacturer and country of origin of the spectrophotometer used should be provided,
line 202 – abbreviation should be explained upon first use,
lines 213, 218, 243 and 311 – there is no supplemental material in the system, so it is difficult to assess,
lines from 229 – do these percentages have to go all the way to the fourth decimal place?
Figure 2 – markers on charts should be enlarged,
lines 305-306 – table title is missing,
lines 465-473 – in my opinion, these data do not add anything interesting to the manuscript.
Author Response
Dear reviewers and editors:
Thank you for your kindly comments and suggestions regarding our manuscript. We have modified the manuscript accordingly, and the amendments were highlighted in yellow in the manuscript. The detailed corrections are listed below point by point:
In the manuscript submitted for review, the authors studied secondary metabolites and antioxidant activities of various edible rosa fruits.
Detailed recommendation:
- line 22 – such marking of the sample will not tell the reader anything,
Response: Thanks for your valuable comment, the abbreviation of sample has been modified accordingly.
- line 22 – abbreviation should be explained upon first use,
Response: Thanks for your valuable comment, the abbreviation has been modified accordingly.
- line 114 – drying condition should be provided,
Response: Thanks for your valuable comment, the drying condition has been provided accordingly.
- Figure 1B – are these data the authors' results? or literature data?
Response: Thanks for your valuable comments, Figure 1B is the authors' results, we have separated the two figures in Figure 1 and placed the Figure 2 (Figure 1B) in the corresponding position of the manuscript.
- line 123 – drying condition should be provided,
Response: Thank you for your valuable feedback. As this sentence overlaps with the sentence in section 2.2, it has been deleted according to the previous reviewer's comments.
- line 126 – concentration temperature should be provided,
Response: Thanks for your valuable comment, the concentration temperature has been provided accordingly.
- line 127 – freeze-drying condition should be provided,
Response: Thanks for your valuable comment, the concentration temperature has been provided accordingly.
- line 131 – instead of rpm, the centrifugal force should be given,
Response: Thanks for your valuable comment, the centrifugal force has been provided accordingly.
- lines 132-137 – website addresses are not needed here,
Response: Thanks for your valuable comment, the website addresses has been deleted accordingly.
- line 157 – typically, a description of the statistical methods used is provided at the end of the methodology section,
Response: Thank you for your valuable advice, we have added a description of the statistical methods in this section accordingly.
- lines 168-180 – it is unclear for what purpose the carbohydrate content was determined, do they have any connection with the antioxidant and bioactive substances detected?
Response: Thanks for your valuable comment, According to previous research reports, polysaccharides from Rosa roxburghii and Rosa laevigata are closely related to their antioxidant effects. Therefore, in order to explore the correlation between antioxidant effects of five species on polysaccharides, we conducted polysaccharide content detection. In the following discussion section, relevant discussions have been conducted.
- line 184 – model, manufacturer and country of origin of the spectrophotometer used should be provided,
Response: Thanks for your valuable comment, the information of microplate reader have been added accordingly.
- line 202 – abbreviation should be explained upon first use,
Response: Thanks for your valuable comment, the explanation of the abbreviations have been added now. All explanation of the abbreviations have been listed on the Appendix.
- lines 213, 218, 243 and 311 – there is no supplemental material in the system, so it is difficult to assess,
Response: Thanks for your valuable comment. Due to software compatibility issues, the supplemental material previously uploaded in the compressed file cannot be opened. Now, the supplemental material have been re uploaded.
- lines from 229 – do these percentages have to go all the way to the fourth decimal place?
Response: Thanks for your valuable comment. The percentages has been adjusted to two decimal places.
- Figure 2 – markers on charts should be enlarged,
Response: Thanks for your kindly comment. but we are very sorry that the markers of the charts was the biggest font size we can do.
- lines 305-306 – table title is missing,
Response: Thanks for your kindly comment. The table title can be found above the table.
- lines 465-473 – in my opinion, these data do not add anything interesting to the manuscript.
Response: Thanks for your kindly comment. This part is the annotation of the of differences in metabolic pathways between Rosa laevigata and R. roxburghii, which can help explain the reasons for the significant differences in metabolites between the two, so we still choose to keep this part.
Thanks for your consideration, and please contact me if you have any questions.
Kind regards,
Wei Gu, PhD
E-mail: guwei2009@126.com
Round 2
Reviewer 1 Report
Comments and Suggestions for Authors
The manuscript has been improved.
However, I suggest numbering the supplementary tables in the order in which they are cited in the text.
Comments on the Quality of English Language
Minor editing of English language required
Author Response
Dear reviewer and editor:
Thank you for your kindly comments and suggestions regarding our manuscript. We have modified the manuscript accordingly, and the amendments were highlighted in yellow in the manuscript. The detailed corrections are listed below point by point:
The manuscript has been improved.
However, I suggest numbering the supplementary tables in the order in which they are cited in the text.
Response: Thanks for your kindly comment, the number of the supplementary tables have been adjusted in the order which they cited in the text.
Thanks for your consideration, and please contact me if you have any questions.
Kind regards,
Wei Gu, PhD
E-mail: guwei2009@126.com